

# The 4.2-ka event, ENSO, and coral-reef development

Lauren T. Toth[1] and Richard B. Aronson[2]

[1]U.S. Geological Survey, St. Petersburg Coastal and Marine Science Center, St. Petersburg, FL, 33701, USA
[2]Department of Ocean Engineering and Marine Sciences, Florida Institute of Technology, Melbourne FL, 32901, USA
*Correspondence to*: Lauren T. Toth (ltoth@usgs.gov)

Variability of sea-surface temperature related to shifts in the mode of the El Niño–Southern Oscillation (ENSO) has been implicated as a possible forcing mechanism for the changes in global-scale, tropical and subtropical precipitation known as the 4.2-ka event. We explore records of coral-reef development and paleoceanography from the tropical

eastern Pacific (TEP) to evaluate the potential impact of the 4.2-ka event on coral reefs. Our goal is to identify the regional climatic and oceanographic drivers of a 2500-year shutdown of vertical reef accretion in the TEP beginning 4.2 ka. The 2500-year hiatus represents ~40% of the Holocene history of reefs in the TEP and was tied to increased variability of ENSO. When ENSO variability abated approximately 1.7–1.6 ka, coral populations recovered and vertical accretion of reef framework resumed apace. The 4.2-ka event appears to have suppressed coral populations

and reef accretion elsewhere in the Pacific Ocean as well. Although the ultimate causality behind the global 4.2-ka event remains elusive, correlations between shifts in ENSO variability and the impacts of the 4.2-ka event suggest that ENSO played a role in climatic changes at that time, at least in the tropical and subtropical Pacific. We outline a framework for testing hypotheses of where and under what conditions ENSO may be expected to have impacted coral-reef environments around 4.2 ka. Although most studies of the 4.2-ka event have focused on terrestrial environments,

we suggest that understanding the event in marine systems may prove to be the key to deciphering its ultimate cause.

## 1 Introduction

The abrupt climatic shift at ~4200 years before present (BP; expressed as years before 1950), known as the 4.2-ka event, is now recognized by many scientists to be the most significant climatic perturbation of the middle to late Holocene (Walker et al., 2012). On 25 June 2018, the International Union of Geological Sciences officially

subdivided the Holocene epoch, with the 4.2-ka event marking the start of the late Holocene, or the Meghalayan age. Although the signature of the 4.2-ka event is not as ubiquitous as the north-Atlantic cooling episode known as the 8.2-ka event, which is the demarcation between the early and middle Holocene, it is now clear that the 4.2-ka event had global climatic, ecological, and cultural impacts (Mayewski et al., 2004; Weiss, 2016). In most locations, the 4.2-ka event was manifested as extreme and abrupt changes in terrestrial hydroclimate (Straubwasser et al., 2003;

Marchant and Hooghiemstra, 2004; Walker et al. 2012), which drove environmental changes (Marchant and Hooghiemstra, 2004; Booth et al., 2005) and cultural collapses (Weiss et al., 1993; Straubwasser et al., 2003; Weiss, 2016; Wu et al., 2016) on a global scale. The ultimate cause(s) of the event and its impact and connection with other regional- to global-scale climatic changes remain unknown (Walker et al., 2012).

Whereas shifts in the climate of polar regions in general, and the northern Atlantic in particular, have been

implicated in the majority of climate events during the late Quaternary (Dansgaard et al., 1993; Mayewski et al., 2004; Booth et al., 2005; Walker et al., 2012), there is little evidence for a high-latitude driver of the 4.2-ka event



(Walker et al., 2012). The event is only weakly recorded in Greenland ice-core records (Johnsen et al., 2001; Marchant and Hooghiemstra, 2004; Mayewski et al., 2004; Rasmussen et al., 2006): major cooling in the northern Atlantic, and the associated onset of northern-hemisphere neoglaciation, preceded the 4.2-ka event by at least 1000

years (Svenden and Mangerud, 1997; Mayewski et al., 2004; Walker et al., 2012; Marcott et al., 2013). The lack of a clear, high-latitude forcing mechanism for the 4.2-ka event has led some researchers to suggest that there may a tropical driver of the climatic changes around this time (Marchant and Hooghiemstra, 2004).

The most significant source of climatic variability in tropical systems is the El Niño–Southern Oscillation (ENSO). Although ENSO originates in the tropical Pacific, it causes global-scale thermal anomalies and changes in

hydroclimate (McPhaden et al., 2006). The climatological manifestations of individual El Niño and La Niña events can vary, but many of the broadscale impacts of ENSO are similar to the global-scale changes inferred to have taken place during the 4.2-ka event. For example, some of the most striking changes associated with the 4.2-ka event were apparent shifts in the strength of the Asian Monsoon (Staubwasser et al., 2003; Wang et al., 2003). ENSO variability has been closely linked with the dynamics of the Asian Monsoon throughout the Holocene (Liu et al. 2000).

Whereas El Niño events are typically associated with a weaker Monsoon and drought conditions, the Monsoon is stronger during La Niña events (Liu et al., 2000; Wang et al., 2003). The similarity in the climatic changes associated with ENSO and the 4.2-ka event, coupled with the evidence for increasing ENSO variability around 4.2 ka (Conroy et al., 2008; Koutavas and Joanides, 2012; Carré et al., 2014), has led a number of researchers to suggest that ENSO may have played a salient role in the 4.2-ka event (e.g., Marchant and Hooghiemstra, 2004; Booth et al.,

2005; Walker et al., 2012). It is not yet clear, however, whether changing ENSO variability around 4.2 ka is likely to be an ultimate driver of broad-scale climatic changes around this time or whether ENSO was a proximal response to other climatic forcing.

Coral reefs are excellent models for elucidating the causes and consequences of climatic excursions such as the 4.2-ka event. Reef-building (hermatypic) corals, whose aragonitic skeletons build the geologic frameworks of

the reefs, are highly sensitive to extremes in temperature and light. As a result, coral reefs are among the most vulnerable ecosystems to modern, anthropogenic climate change (e.g., Hoegh-Guldberg, 1999). Photosynthetic dinoflagellates, called zooxanthellae, which live endosymbiotically within the cnidarian host, provide the animal component of the coral holobiont (the entire animal–plant–microbial symbiosis) with nearly all of its energetic needs (Muscatine and Porter, 1977). Persistently high temperature and light levels, which are now occurring with

increasing frequency as a result of anthropogenically-driven climate change, disrupt the mutualistic relationship between the coral animal and the zooxanthellae, leading to coral bleaching and associated morbidity and mortality (Fig. 1; Brown, 1987; Glynn, 1991; Hoegh-Guldberg, 1999; Donner et al., 2005; Anthony, 2016; and many others). Cold-water conditions are also inimical to corals (Ginsburg and Shinn, 1994; Precht and Aronson, 2004; Lirman et al., 2011; Kemp et al., 2016; Toth et al., 2018). Corals are vulnerable to a variety of other stresses and disturbances,

including sedimentation, subaerial exposure, nutrient loading, hurricane damage, low-pH conditions, and low salinity (Kleypas et al., 1999; Aronson and Precht, 2001; Buddemeier et al., 2004; Goldberg, 2013), which can increase as a result of (or in tandem with) climatic shifts in one direction or another. Reef frameworks and the coral



skeletons that constitute them preserve well in the fossil and subfossil record, providing a historical record of the response of coral reefs to climate change (Jackson, 1992). Moreover, many of the perturbations listed above leave

geochemical and taphonomic traces that can be used as proxies to reconstruct changes in environmental conditions (Flannery et al. 2018; Cobb et al., 2013; Toth et al., 2015a, 2015b).

Whereas the majority of the records of the 4.2-ka event have come from terrestrial environments, the contemporary impacts of ENSO are felt most keenly in marine ecosystems. Understanding whether and how marine ecosystems responded to climatic changes around 4.2 ka is, therefore, critical to deciphering the ultimate drivers of

the 4.2-ka event. Here we explore paleoecological and paleoceanographic records from marine environments in the tropical Pacific. We focus on the long-term collapse of coral-reef development in the tropical eastern Pacific to evaluate the role of ENSO in the 4.2-ka event.

## 2 Climate and coral-reef development in the eastern tropical Pacific

### 2.1 Ecology of Coral Reefs in the Eastern Tropical Pacific

Holocene reefs of the TEP are for the most part small and poorly developed, as is generally the case for reefs on the eastern margins of ocean basins (Darwin, 1862; Cortés, 1993).  Regional and local diversities of hard corals are low in the TEP. Several living reefs, however, overlie well-developed Holocene frameworks that preserve millennial-scale records of regional and larger-scale climatic variability, and the responses of coral assemblages to that variability.

The shallowest habitats of contemporary reefs off the Pacific coast of Panamá, ranging from 1–5 m water depth, are characterized by fields of branching corals of the genus *Pocillopora* (Fig. 1a; Glynn and Macintyre, 1977; Cortés, 1997; Glynn and Ault, 2000). The stands of *Pocillopora* are dominated by *P. damicornis*, but they include several other species in the genus. Massive corals, primarily *Pavona* spp., *Porites* spp., and *Gardineroceris planulata*, become more abundant with increasing depth below the zone of essentially monogeneric occupation by

*Pocillopora*.

Richmond (1987, 1990) suggested that *P. damicornis* in the TEP had adopted a life-history strategy of exclusive, or nearly-exclusive, asexual reproduction. In this scenario, the populations were seeded by sexually produced planula larvae from the central Pacific, which were transported by the North-Equatorial Countercurrent and by eastward equatorial flow during El Niño conditions (Glynn and Ault, 2000). Wood et al. (2016), however,

cast considerable doubt on Richmond's proposed oceanographic teleconnection between the central and eastern Pacific. Furthermore, genetic analysis by Combosch et al. (2008) demonstrated that the eastern-Pacific populations of *Pocillopora* reproduce sexually. The genetics showed a difference in reproductive strategy in the genus *Pocillopora* between the eastern Pacific and the rest of its range in the Indo-Pacific: *Pocillopora* reproduce sexually by broadcast-spawning in the eastern Pacific, whereas they brood internally-fertilized or parthenogenetic eggs in

central and western Pacific, and in the Indian Ocean.

Broadcast-spawning is correlated with interspecific hybridization among the species of *Pocillopora* in the TEP. Hybridization in the TEP is in turn associated with a radically different functional ecology of (hybridized) *P.*





*damicornis* in the eastern Pacific, including a far more robust skeletal morphology, ecological dominance, and the construction of reef-frameworks composed predominantly of *Pocillopora* branches. Within the eastern Pacific,

limited gene flow among populations of (hybridized) *P. damicornis* (henceforth simply '*P. damicornis*' or simply '*Pocillopora*') raises the possibility of adaptation to regional-scale oceanographic gradients of upwelling (Combosch and Vollmer, 2011).

Ecological observations prior to 1982 suggested that dense stands of *P. damicornis* had persisted throughout the TEP for decades at least. Push-cores extracted from reef-frameworks in both the Gulf of Panamá, where seasonal

upwelling is strong, and the Gulf of Chiriquí, where seasonal upwelling is weak or absent, appear to corroborate the ecological data, suggesting continuous dominance of *P. damicornis* for at least the last 1500 yr (Toth et al., 2012, 2017). These ecological and paleobiological lines of evidence are not strictly comparable, however, because the temporal precision of the cores is probably not great enough to detect decadal-scale events. The cores could belie short periods of interrupted coral growth.

**2.2 Response to ENSO Events**

Contemporary populations of corals on reefs of the TEP highlight the responses of coral populations to sequential perturbations by ENSO. The 1982–83 El Niño was one of the strongest to affect the eastern Pacific in recent history (Glynn, 1988a; Glynn et al., 2001). (The impacts of this event were less severe elsewhere in the tropical Pacific, however.) Persistently high water temperatures for 14 months resulted in severe, regional-scale bleaching of corals

(Fig. 1b) and other cnidarians bearing endosymbiotic algae (Wellington and Glynn, 2007). Coral mortality ranged from 50% in Costa Rica to 97% in the Galápagos Islands and correlated with the degree of thermal stress the reefs experienced in each location (Glynn, 1990). Mortality levels were intermediate off the Pacific coast of Panamá. In the Gulf of Panamá (upwelling), coral mortality was 85%. In the adjacent Gulf of Chiriquí (non-upwelling), mortality was similar, averaging 76% (Glynn, 1990, 1991; Glynn and Colgan, 1992; Wellington and Glynn, 2007).

*Pocillopora damicornis* had been the dominant benthic component in shallow reef habitats prior to the 1982–83 event, covering up to 90% of the available substratum at 3–5 m depth (Fig. 1a). *Pocillopora* proved especially vulnerable to bleaching-induced mortality and vast fields were killed, although mortality was variable on multiple spatial scales (Glynn, 1990; Macintyre and Glynn, 1990). A collateral effect of *Pocillopora* mortality was the decline of symbiotic crustaceans that defend the corals against attack by the predatory crown-of-thorns starfish,

*Acanthaster planci*. At Uva Island in the Gulf of Chiriquí, *Acanthaster* were able to crawl across freshly killed thickets of *Pocillopora* to attack and kill massive corals that had not bleached as severely and did not harbor the crustaceans (Glynn, 1985, 1990). *Acanthaster* did not then and does not now occur in the Gulf of Panamá (Glynn, 2004).

Algal turf growing on the dead-coral surfaces enhanced populations of regular echinoids by increasing their

food supplies. Higher densities of grazing sea urchins greatly increased bioerosion and suppressed coral recruitment, resulting in a switch from net vertical reef accretion to net framework erosion (Glynn, 1988a, 1988b, 1990; Glynn, and Colgan 1992; Eakin, 1996; Reaka-Kudla et al., 1996). Centennial-scale increases in the frequency of severe



ENSO events such as the one in 1982–83 could be responsible for the poor development of reefs observed in many areas of the eastern Pacific (Glynn and Colgan, 1992; Glynn, 2000).

A second extreme ENSO event killed an estimated 16% of corals worldwide in 1997–98 (Wilkinson, 2000). The 1997–98 and 1982–83 events were of approximately equal magnitude and duration in the eastern Pacific. Both events were enhanced by global warming and they were the two most intense events in the preceding 50 yr (Hansen, et al. 1999; Karl et al., 2000; Enfield, 2001).

    Coral mortality in the eastern Pacific was lower in 1997–98 (Guzmán and Cortés, 2001; Vargas-Ángel et al., 150 2001). There was essentially no mortality in the Gulf of Panamá because, whereas seasonal upwelling was suppressed during the 1982–83 El Niño event, upwelling was unaltered in 1997–98 and cooled the water column (Riegl and Piller, 2003; Glynn et al., 2001). Muted levels of coral mortality elsewhere, including only 13% mortality in the Gulf of Chiriquí, were correlated with the presence of thermally resistant zooxanthellae in the corals (Glynn et al., 2001; Baker et al., 2004; see also D'Croz and Maté, 2004). Collateral effects following the 1982–83 event, 155 particularly enhanced predation by *Acanthaster* (in the Gulf of Chiriquí) and increased bioerosion by sea urchins, were also far less severe after the 1997–98 ENSO, due to precipitous declines of both taxa (Eakin, 2001; Glynn et al., 2001; see also Fong and Glynn, 2001). The effects of the 2015–16 El Niño event were also minimal in Pacific Panamá. Upwelling again buffered the reefs in the Gulf of Panamá from elevated temperatures, and most reefs in the Gulf of Chiriquí experienced coral bleaching but only minor bleaching-related mortality.

Coral populations have recovered to varying degrees from the El Niño events in 1982–83 and 1997–98, having been seeded by populations that persisted within refugia in the eastern Pacific (Wood et al., 2016). Coral cover declined in the Gulf of Chiriquí at 3–7 m depth after 1983. By 2002, coral cover was 12% at Uva Island (down significantly from 35%) and 8% at Secas Island (down non-significantly from 11%; Wellington and Glynn, 2007). Coral cover at Saboga Island in the Gulf of Panamá recovered at 2–3 m depth from 0% in 1984 to the pre-ENSO 165 level of 50% by 1992. At 3–5 m depth, some areas of the Saboga reef recovered to 50% within a few years, but large areas of coral rubble have persisted to this day (Fig. 1c), and this has also been the case on many other reefs. The areas of coral rubble consist primarily of taphonomically degraded *Pocillopora* branches, which are covered in algal turfs, encrusted by coralline algae, and colonized at low frequency by the early-successional coral *Psammocora stellata*.

The foregoing brief discussion summarizes the responses of corals and reef communities of the tropical eastern Pacific to repeated ENSO events. Food webs on eastern Pacific reefs are complex despite their low diversity (Glynn, 2004). Corallivores other than *Acanthaster*, including pufferfish (Tetraodontidae) and gastropods (Pediculariidae and Muricidae), have effects that vary on multiple scales. On the other hand, because it is situated close to the equator, Panamá has not experienced a hurricane since at least as early as 1871 (Neumann et al., 1999), removing 175 hurricanes as a factor (but not storms and wave action in the winter dry season).

## 2.3 The Question of Prior Occurrence

The TEP has been affected by more than 30 strong ENSO events during the past 500 yr and, perhaps, hundreds of events as strong as the 1982–83 and 1997–98 ENSOs during the Holocene (Colgan, 1990; Moy et al., 2002; Rein et



al., 2005). Cores drilled from massive coral colonies (coral heads) throughout the Indo-Pacific display a trend
toward more intense ENSO events after 1976 (Urban et al., 2000); however, although ENSO variability during the
last century may be significantly higher than many times in the past, there is precedent in the fossil record for the
intensity of recent events (Cobb et al., 2013). Whereas gross patterns of reef development have been examined in the
eastern Pacific (Glynn and Macintyre, 1977; Macintyre et al., 1992; Cortés et al., 1994), biotic turnover and its
geological consequences remain understudied.

At least some effects of the 1982–83 ENSO event were unprecedented on a scale of centuries. First, colonies of
the massive corals *Porites lobata* and *Pavona clavus* in the Galápagos were 350–425 years old at the time of their
death during the event (Glynn, 1990). Second, Dunbar et al. (1994) examined a colony of *Pavona clavus* that was
part of the reef community at Urvina Bay in the Galápagos. In 1954, Urvina Bay was tectonically uplifted more than
7 m, preserving in excellent detail the effects of a strong El Niño in 1941 (Colgan, 1990). The *Pavona* colony, which
had survived the 1941 event, showed more than 350 yr of continuous growth before its death in the 1954 uplift.
Clearly, some massive corals had lived through previous, strong ENSO events. On the other hand, partial colony
mortality and regeneration of the massive species *G. planulata* at Uva Island marked both the 1982–83 and 1997–98
ENSOs, so even the latter event was not without ecological consequences (Wellington and Glynn, 2007).

More germane are Colgan's (1990) observations on uplifted stands of *Pocillopora* at Urvina Bay. Much of the
*Pocillopora* was in poor taphonomic condition (but excellently preserved in that poor condition; Colgan and
Malmquist, 1987), undoubtedly the result of 13 yr of bioerosion by urchin grazing after the ENSO of 1941 and prior
to the uplift. The uplifted reef contained clear evidence of bioerosive undercutting and collapse of the *Pocillopora*
framework caused by sea urchins, *Eucidaris galapagensis*, quite similar to what was observed in living communities
in the Galápagos after the 1982–83 event. Furthermore, *Eucidaris* tests were scattered in large numbers at the base
of the eroded *Pocillopora* framework.

The question of prior occurrence is important because, as discussed above, repeated El Niño events could
explain why reefs are poorly developed in the eastern Pacific. Can we identify mass coral-kills and excursions from
the *Pocillopora*-dominated state in the subfossil record of eastern Pacific reefs? Colgan's (1990) results and post-
1983 observations throughout the region imply that under the right conditions it should be possible to detect
bioeroded horizons of *Pocillopora* and shifts in dominance between coral species.

*Pocillopora*-dominated reefs are not well-developed in the eastern Pacific due to the narrow shelves on which
they grow; exposure in at least some locations to cold and acidic, upwelling waters; and episodic bleaching during
ENSO events (Cortés et al., 1994; Glynn and Maté, 1997; Kleypas et al., 2006). Glynn and Colgan (1992; Glynn,
2000) argued that eastern Pacific reefs accrete poorly because the corals are periodically killed by severe El Niño
events and then bioeroded by echinoids. Our study of the geologic record, in contrast, point to a protracted phase
shift, during which the vertical accretion of Holocene reef-frameworks off the Pacific coast of Panamá was
suppressed for far longer than the return time of strong ENSO events.

Cores we extracted from the uncemented frameworks of three Panamanian reefs across a gradient of upwelling,
and dated with radiocarbon and uranium-series techniques, showed that vertical accretion essentially ceased from
~4100 to 1600 cal BP (Fig. 2; calibrated calendar years before 1950; Toth et al., 2012). The hiatus in growth lasted



approximately 2500 years, meaning the reefs were in a phase of negligible growth for as much as 40% of their history (Toth et al., 2012).

In each core, active coral growth was represented in the cores as long intervals of well-preserved *Pocillopora* branch-fragments. The hiatus was manifested as a thin layer of taphonomically degraded *Pocillopora* branch-
fragments, coralline algae, and *Psammocora stellata*, all of which characterize ENSO-generated rubble-fields on contemporary reefs in Panamá. During the rest of their history, however, the reefs grew upward at rates comparable with the vertical-accretion rates of Caribbean reefs (Toth et al., 2012). The 'poor' Holocene development of these reefs is, therefore, a consequence of the hiatus. Although ENSO suppresses populations of *Pocillopora* on subdecadal to multidecadal scales, that has not been the cause of low bulk-accretion rates through the Holocene, as
previously suspected. The tempo and causality of reef development in Pacific Panamá represent a vastly scaled-up version of the earlier model of Glynn (2000).

**3 ENSO and Coral-Reef Collapse at 4.2 ka**

There is now broad consensus across both climate models and paleoclimate reconstructions that the middle Holocene was a period of relatively low ENSO variability (Clement et al., 1999; Sandweiss et al., 2001; Rein et al.,
2005; Koutavas et al., 2006; Cobb et al., 2013; Carré et al., 2014; Liu et al., 2014; Emile-Geay et al., 2016; Leonard et al., 2016b; Thompson et al., 2017). In particular, the interval 5–3 ka stands out as a period when paleoclimate records from throughout the tropical Pacific show a significant reduction in ENSO activity (Emile-Geay et al., 2016). Although the exact timing of the middle-Holocene ENSO minimum varies among reconstructions, some studies have detected exceptionally low ENSO variability between 4.3 and 4.2 ka (Cobb et al., 2013; McGregor et
al., 2013; Leonard et al., 2016b). Somewhat paradoxically, ~4.2 ka also appears to be a point of inflection in most ENSO records, after which time ENSO variability increased, eventually leading to the establishment of the modern ENSO regime (Conroy et al., 2008; Koutavas and Joanides, 2012; Carré et al., 2014).

ENSO appears to be a primary control on coral growth and reef development in the TEP over scales of decades to millennia (Glynn and Colgan, 1992; Glynn et al., 2001; Toth et al., 2012, 2017). In Pacific Panamá, the
elevated water temperatures and high irradiance (low cloud cover) associated with the strong El Niño events in 1982–83 and 1997–98 caused widespread coral mortality (Glynn et al., 2001). La Niña is also problematic for reefs in Pacific Panamá, as lowered sea level in the TEP during La Niña events causes more frequent coral mortality associated with sub-aerial exposure (Eakin and Glynn, 1996; Toth et al., 2017). In addition, La Niña is associated with elevated rainfall in Pacific Panamá, which increases turbidity, and enhanced upwelling, both of which can
suppress coral growth (Glynn, 1976). The dominant role of ENSO in modulating modern reef development in the TEP led us to hypothesize that ENSO variability may have also shaped the dynamics of Panamanian reefs over millennial timescales.

We tested the hypothesis that changes in ENSO activity around ~4.2 ka triggered reef shutdown in Pacific Panamá by evaluating geochemical proxy records from corals in our cores (Fig. 2; Toth et al., 2015a, 2015b).
Although we do not have any records at precisely 4.2 ka, our reconstructions give insights into the mean climatic and oceanographic states bounding 4.2 ka, which we evaluate in the context of records of ENSO variability from



throughout the tropical Pacific (Fig. 2). Our record of oceanic radiocarbon variability, which provides a proxy for ocean circulation, suggests that upwelling was enhanced beginning at 4.3 ka, and continued to intensify over at least the next 500 years (Fig. 2a; Toth et al., 2015b). Paleoclimate reconstructions, which we based on Sr/Ca and $\delta^{18}O$ in

the coral skeletons, suggest that these oceanographic changes were followed by an abrupt transition to a cooler, wetter climate beginning by ~3.9 ka (Fig. 2b,c; Toth et al., 2015a). A lake record from the Galápagos recorded a contemporaneous period of anomalous drying, which in that location suggests enhanced La Niña activity (Fig. 3; Conroy et al., 2008). Together, those inferred conditions indicate that an enhanced La Niña-like climate regime was established just after 4.2 ka, which likely provided the initial trigger that shutdown reef development in Pacific

Panamá for the next 2500 years (Toth et al., 2012, 2015a).

       Whereas the period from 5–3 ka may have been a time of low ENSO variability *on average* (Emile-Geay et al., 2016), the time just after 4.2 ka was likely a time of high climatic volatility in the tropical Pacific (Fig. 3). At least one record indicates that a multi-decadal period of enhanced ENSO variability followed just 500 years after the putative low in ENSO activity at ~4.2 ka (Corrège et al., 2000). Our data from Pacific Panamá suggest that this

period would have been followed by an abrupt transition to La Niña-like conditions by 3.9 ka (Toth et al., 2015a, 2015b), and a recent study suggests that La Niña events were also more frequent from 3.5–3 ka (Thompson et al., 2017). A record of El Niño-related flooding from Peru implies that the strongest El Niño events of the Holocene occurred 4–2 ka (Rein et al., 2005), and a series of records from the same region suggest a possible peak in the frequency of El Niño events around 3 ka (Sandweiss et al., 2001; Moy et al., 2002). The beginning of the late

Holocene was also characterized by a period of high variability in the position of the Intertropical Convergence Zone (ITCZ) supporting the idea that regional climatic variability was high at this time (Haug et al., 2001). These studies support the conclusion that ENSO variability was enhanced after 4.2 ka (Conroy et al., 2008; Koutavas and Joanides, 2012; Carré et al., 2014).

       Stronger and/or more frequent El Niño and La Niña events would have acted to suppress reef development

after 4.2 ka (Toth et al., 2012). ENSO variability continued to increase after the end of the hiatus in reef development (Conroy et al., 2008; Cobb et al., 2013; Thompson et al., 2017); however, major changes in the mode of ENSO coincided with its termination (Fig. 3). The variability in the mean position of the ITCZ had declined by ~2.5 ka (Haug et al., 2001). Although ENSO variability and the absolute number of El Niño events may have increased beginning 2 ka (Sandweiss et al., 2001; McGregor and Gagan, 2004; Conroy et al., 2008; Thompson et al.,

2017), during the hiatus, El Niño events were likely stronger (Rein et al., 2005) and La Niña event may have been more frequent (Thompson et al., 2017). We hypothesize that the waning influence of La Niña and reduction in El Niño strength permitted accretion to resume between ~1.8–1.5 ka (Toth et al., 2012, 2015a).

       We conclude that ENSO was likely the primary driver of the collapse of coral reefs in Pacific Panamá and, perhaps, other locations around the tropical Pacific at ~4.2 ka. The potential role of changing ENSO variability in

ecological changes elsewhere at this time is less clear, but there is some evidence to support the idea (Li et al., 2018). Furthermore, even if enhanced ENSO variability triggered reef-shutdown 4.2 ka, that does not mean ENSO-related changes necessarily accounted for the next 2500 years of suppressed reef development; Fig. 2 implies a



concatenation of conditions that might or might not have been dependent or semi-independent. Some evidence does in fact point to climatic shifts around the time of the 4.2-ka event that persisted for millennia (Selvaraj et al. 2008).

There is still considerable disagreement about timing and magnitude of the putative shifts in ENSO variability during the middle to late Holocene (Emile-Geay et al., 2016). One problem in linking ENSO as a driver or response to the 4.2-ka event is the asymmetry in the ENSO signal between the eastern and central Pacific that can be observed from both recent El Niño events (Fig. 4; Takahaski et al., 2011) and in the Holocene record (Carré et al., 2014; Karamperidou et al., 2015). Indeed, there appear to be two distinct "flavors" of ENSO based on the

location of the strongest thermal anomalies—eastern Pacific and central Pacific events—and the dominance of these ENSO modes may have varied over millennial timescales (Karamperidou et al., 2015). Another problem is the large temporal uncertainties associated with many of the existing paleoclimate records of ENSO. Currently, it is not possible to resolve clearly the centennial-scale shifts in ENSO variability around 4.2 ka. Reducing the temporal uncertainties in paleo-ENSO records at 4.2 ka and the dominant flavor of ENSO at this time, by targeting ENSO

proxies from throughout the tropical Pacific, will be a critical first step to determining whether shifts in ENSO variability can be linked to the 4.2-ka event on a geographic scale and the subsequent 2.5 millennia of impacts on eastern Pacific reefs.

**Regional- to Global-Scale Impacts of ENSO and the 4.2 ka Event**

ENSO is the most prominent interannual driver of global-scale climate variability. Although it originates in the

tropical Pacific, it impacts regional climates around the world (McPhaden et al., 2006). The global impacts of ENSO are primarily a result of the influence of ENSO-based sea-surface temperature anomalies on other drivers of climate variability such as the ITCZ, the Asian Monsoon, and the North Atlantic Oscillation (NAO). For example, the warm (cool) tropical Pacific sea-surface temperatures associated with El Niño (La Niña) generally drive a southerly (northerly) shift in the mean position of the ITCZ, which results in changes in precipitation from the tropical Pacific

to the neotropics, to Africa (Haug et al., 2001; Chiang et al., 2002; Sachs et al., 2009). Similarly, the Asian monsoon is generally suppressed during El Niño events and enhanced during La Niña events (Liu et al., 2000; Wang et al., 2003). Although the relationship between ENSO and the NAO, which has a strong influence on European climates, is less straightforward, the NAO is typically negative (positive) during El Niño (La Niña) events (Rodríguez-Foncesca et al., 2016).

The global impacts of El Niño and La Niña events can vary significantly from event to event; however, there are striking spatial correlations between the global signature of ENSO and that of the 4.2-ka event (summarized in Davey et al., 2013). In the majority of the records to date, the 4.2-ka event has been characterized by changes in terrestrial hydroclimate (Weiss, 2016). Whereas the event was characterized by widespread aridification in southern Europe, the Middle East, northern Africa, midcontinental North America, and parts of eastern and

southwest Asia (Weiss et al., 1993; Straubwasser et al., 2003; Marchant and Hooghiemstra, 2004; Booth et al., 2005), other locations, such as western South America, experienced marked increases in precipitation (Marchant and Hooghiemstra, 2004). With the exception of Europe, which is weakly teleconnected with ENSO, the mean shifts in precipitation that occurred during the 4.2-ka event are generally consistent with those observed during a typical El





Niño event (Davey et al., 2013). The changes in hydroclimate during typical El Niño events even replicate the zonal

gradient in precipitation observed across northern South America at ~4.2 ka (Marchant and Hooghiemstra, 2004; McPhaden et al., 2006; Davey et al., 2013).

Like many records of ENSO variability from the tropical Pacific, our records from Pacific Panamá do not display a sudden change in El Niño activity at 4.2 ka (Toth et al., 2015a; Emile-Geay et al., 2016); however, most reconstructions of Holocene ENSO suggest that there was a transition to increasing ENSO variability around this

time (Conroy et al., 2008; Koutavas and Joanides, 2012; Cobb et al., 2013; Carré et al., 2014). Whereas the 4.2-ka event was manifested as an abrupt, short-lived climatic event in many locations, the ecosystem response to the event may have been more gradual and more protracted. In Pacific Panamá, the onset of reef collapse in the tropical eastern Pacific just after 4.2 ka was associated with a transition to a cooler, La Niña-like climate (Toth et al., 2015; see also Cabarcos et al., 2014), which triggered a regime shift that lasted for millennia. Similarly, there is evidence

that the onset of aridification in the mid-continent United States was followed by glacial advances in western Canada, from 4.2–3.8 ka, associated with a cooler, wetter climate (Menounos et al., 2008; Mayewksi et al., 2004) that is characteristic of La Niña conditions in that region (Davey et al., 2013). Thus, whereas many of the abrupt changes in terrestrial hydroclimate during the 4.2-ka event are similar to those observed during El Niño events, the ecosystem changes following the 4.2-ka event may reflect responses to secondary changes in regional climatic

systems.

We are not arguing here that ENSO was necessarily the ultimate cause of the 4.2-ka event; however, the shift in ENSO variability after 4.2 ka and the similarity between the global impacts of ENSO and the global footprint of the 4.2-ka event suggest a strong connection. The climatic forcing that produced the changes in ENSO variability during the middle Holocene are still being debated, but it is generally thought that changes in ENSO during the Holocene

have been driven by gradual changes in the climate system associated with changing insolation and/or feedbacks with annual climate cycles (McPhaden et al., 2006; McGregor et al., 2013; Emile-Geay et al., 2016;). For example, several investigators (Koutavas et al., 2006; McGregor et al., 2013) have suggested that the weaker ENSO variability observed in the central Pacific at ~4.3 ka was related to the more northerly ITCZ at this time, which would have also enhanced eastern-Pacific upwelling, produced cooler sea temperatures in the TEP, and driven a

stronger zonal SST gradient that suppressed ENSO (Clement et al., 2000). In the absence of a clear high-latitude driver for the 4.2-ka event, these changes in the tropical ocean must be considered as likely contributors to the 4.2-ka event (Marchant and Hooghiemstra, 2004).

**Prospectus**

Because the global teleconnections of El Niño and La Niña can vary significantly from event to event (Fig. 4;

McPhaden et al., 2006), ecosystems of the tropical and subtropical Pacific are likeliest to yield the best evidence for evaluating the potential role of ENSO in 4.2-ka event. Corals and coral-reef communities that were living under chronically stressed conditions should have been particularly vulnerable to increased ENSO variability 4.2 ka. If the hiatus was triggered by La Niña-like conditions (Fig. 4), we would predict that the 4.2 ka event would be most strongly manifested in reef records from areas of upwelling and in seasonally cooler, subtropical environments



(compared with less-seasonal, tropical environments). Nearshore habitats, which are more influenced by the thermal
swings and high sedimentation rates associated with terrigenous input, should be more vulnerable, compared with
reefs exposed to clearer oceanic waters. Reefs off the Pacific coast of Panamá and Costa Rica, which are exposed to
a strong terrigenous influence and strong upwelling (in some areas), display the hiatus (Table 1; Cortés et al., 1994;
Toth et al., 2012). Latitudinally marginal lagoonal habitats elsewhere in the Pacific should also manifest the hiatus

(see Cruz et al., 2018). Preliminary evidence corroborates that hypothesis in the western Pacific, in turbid, lagoonal
habitats south of the Great Barrier Reef, in southern China, and in Japan (Table 1; Lybolt et al., 2011, Hamanaka et
al., 2012; Yamano et al., 2012; Xu et al., 2018). Whether La Niña-driven reef shutdown occurred broadly on
subtropical reefs of the central Pacific remains unknown, but there is some evidence that increased wave energy tied
to ENSO suppressed vertical reef accretion in Hawaii around the time of the 4.2-ka event (Table 1; Grossman and

Fletcher, 2004; Rooney et al., 2004).

The impacts of increased El Niño intensity (Fig. 4), by contrast, might have been attenuated in lagoonal and
nearshore habitats, where turbidity associated with terrigenous runoff from increased precipitation (in many
locations, though not in Panamá) would have decreased light penetration and thereby offered a measure of
protection from bleaching (Perry et al., 2012; Cacciapaglia and van Woesik, 2015). Above some threshold

amplitude, however, the ENSO swings would have overwhelmed the capacity of even turbid-water reefs to resist
bleaching (e.g., Aronson et al., 2002). Furthermore, El Niño conditions are accompanied by increased precipitation
in most places, with potentially negative impacts on corals.

Across the equatorial Pacific, contemporary corals and coral reefs at longitudes that have experienced higher
temperatures over the previous century have proven more resistant to bleaching events. The eastern and central

Pacific, including tropical as well as subtropical areas, have thus been more vulnerable for historical reasons than the
western Pacific (Thompson and van Woesik, 2009). If the near-term geographic patterns of ENSO history are
indicative of patterns earlier in the Holocene, then on the basis of El Niño-type impacts we would expect the 4.2-ka
event to have manifested strongly in the eastern and central Pacific, and less so in the western Pacific. This
hypothesis could be tested by coring Holocene reef-frameworks near the equator on a longitudinal gradient across

the Pacific. The pre-existing marginality of individual reefs, as a result of additional sources of environmental
variability (such as upwelling) and their ecological characteristics (including their species diversity), would have
modulated their resilience to El Niño and the likelihood that El Niño disturbances would have left a signature in
their fossil record.

Cobb et al. (2013) documented low ENSO variability 4.2 ka at Kiritimati in the Line Islands of the central

Pacific, which appears opposite to the trajectory of ENSO variability in Pacific Panamá at that time. The results
from Kiritimati potentially falsify our hypothesis of an event at 4.2 ka manifested in reef frameworks of the central
Pacific, as well as the idea that contemporary geographic patterns of ENSO variability reflect patterns that prevailed
4.2 ka. Parsing the spatially variable impacts of ENSO and other, more localized drivers such as relative sea level
(Leonard et al., 2016a, 2016b) to explain patterns of reef development will be a significant challenge going forward.

The preponderance of research to date on the causes and consequences of the 4.2-ka event has focused on
terrestrial environments and cultural impacts. In this paper we have shown that there were significant perturbations





to coral-reef environments in Pacific Panamá, and throughout the tropical Pacific, around 4.2 ka. Without a clear high-latitude driver for the 4.2-ka event, it stands to reason that the climatic changes around this time could have originated in the tropical oceans and may have been related to contemporaneous changes in ENSO variability.

Coral-reef archives may provide the key to discerning the role of ENSO in the 4.2-ka event.

**Acknowledgements**

This paper grew out of an international workshop, 'The 4.2 ka BP Event,' which was organized by G Zanchetta, H. Weiss, and M. Bini and held in the Dipartimento di Scienzi della Terra at the University of Pisa in January 2018. We thank the organizers for the opportunity to participate in the workshop. R. Bradley, N. Graham, W. Precht, R.

van Woesik, M. Bush, and J. Richey provided valuable advice and discussion. Our research was supported by grant OCE-1535007 from the U.S. National Science Foundation. Toth's research is also funded by the Coastal and Marine Hazards and Resources Program of the U.S. Geological Survey. This is contribution number XXX from the Institute for Global Ecology at the Florida Institute of Technology. Any use of trade, firm, or product names is for descriptive purposes only and does not imply endorsement by the U.S. Government.

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



**Table 1: Summary of the records of perturbations to coral-reef ecosystems around the time of the 4.2-ka event. Reported radiocarbon ages were calibrated with the marine radiocarbon curve and regional marine reservoir age corrections, using CALIB software (v.7.0.2; Stuiver et al., 2018).**

| Type of impact | Location | Time period (ky BP) | Reference |
|---|---|---|---|
| Hiatus in reef development | **Pacific Panamá** | | |
| | *Contadora* | 4.3–1.5 | Toth et al. 2012 |
| | *Iguana* | 4.2–1.6 | Toth et al. 2012 |
| | *Canales de Tierra* | 4.0–1.8 | Toth et al. 2012 |
| | **Costa Rica** | | |
| | *Golfo Dulce* | 4.6–2.0 | Cortés et. al 1994 |
| | **Great Barrier Reef** | | |
| | *Moreton Bay* | 4.2–2.0 | Lybolt et al. 2011 |
| | **Japan** | | |
| | *Kodakara Is.* | 4.4–4.0 | Hamanaka et al. 2012 |
| Shutdown of reef development | **Hawaii** | | |
| | *Kailua, O'ahu* | 5.0–present | Rooney et al. 2004 |
| | *Punalu'u, O'ahu* | 4.9–present | Rooney et al. 2004 |
| | *Hale o Lono, Moloka'i* | 4.8–present | Rooney et al. 2004 |
| | *Mana, Kaua'i* | 4.5–0 | Rooney et al. 2004 |
| Coral mortality | **China** | | |
| | *South China Sea* | 4.2–3.8 | Xu et al. 2018 |
| Decline in coral $\delta^{13}C$ | **Pacific Panamá** | | |
| | *Contadora* | 3.9–3.6 | Toth et al. 2015 |
| | **China** | | |
| | *South China Sea* | 3.8 | Xu et al. 2018 |





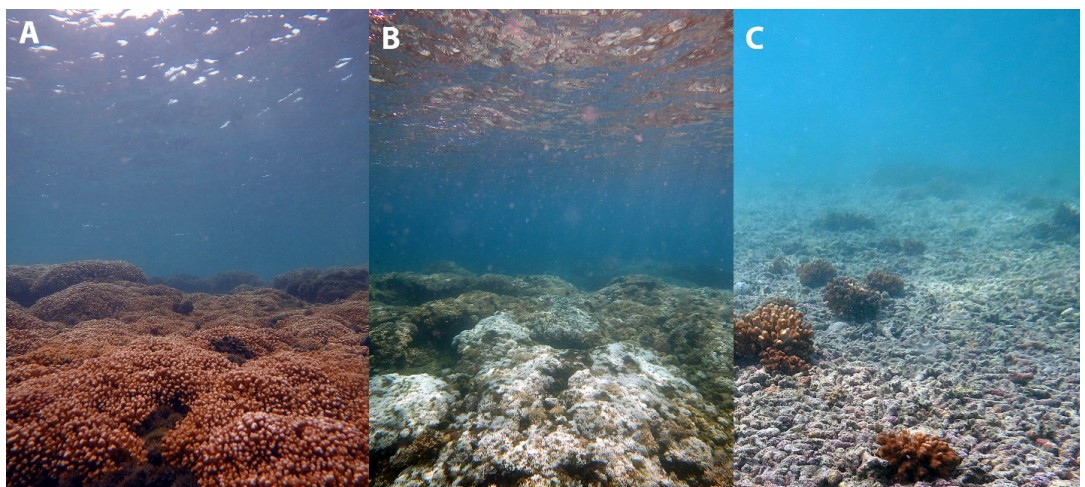


**Figure 1: Photographs depicting the typical states of coral reefs in Pacific Panamá (A) before, (B) during, and (C) after an El Niño event. (A) shows an example of a well-developed coral reef dominated by branching *Pocillopora* corals. (B) shows a reef dominated by *Pocillopora* experiencing high levels of coral bleaching during the 2015–16 El Niño event. (C) shows a large area of *Pocillopora* rubble that has persisted since the**

**1982–83 El Niño event.**





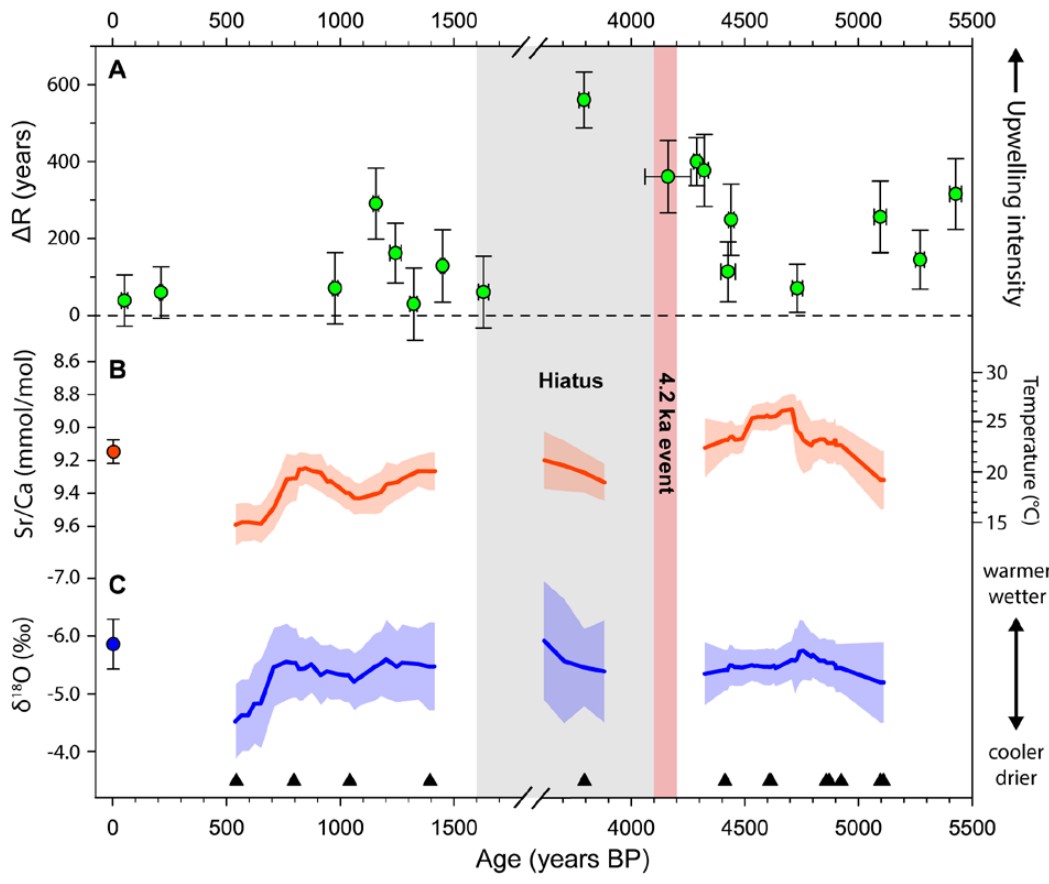

**Figure 2. Coral-based reconstructions of oceanographic and climatic variability in Pacific Panamá from 5500
BP to present (after Toth et al., 2015a,b). (A) provides a reconstruction of upwelling intensity in the Gulf of
Panamá based on the location radiocarbon reservoir age offset, ΔR (Toth et al., 2015b). (B & C) provide
reconstructions of temperature and salinity variability for Pacific Panamá based on Sr/Ca and δ¹⁸O of corals
sampled from a core collected from the reef at Contadora Island in the Gulf of Panama. The timing of the 4.2
ka event and the hiatus in the development of reefs in Pacific Panamá are indicated by the red and gray bars,
respectively.**





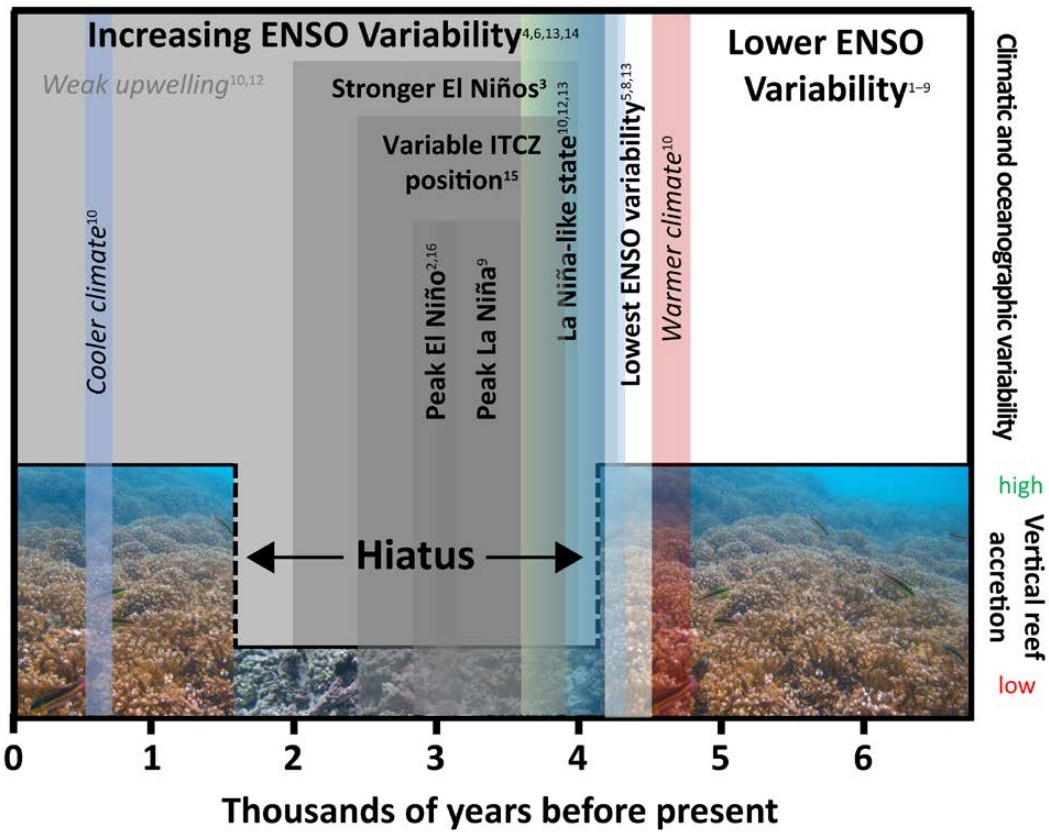

**Figure 3: Schematic diagram of the reconstructed changes in ENSO variability in the tropical Pacific in relation to vertical reef-accretion in Pacific Panamá. Colored bars are based on a palaeoceanographic reconstruction from Pacific Panamá shown in Figure 2; gray bars summarize the results of other paleoclimate studies from throughout the tropical Pacific. Superscripts reference the studies support the climatic and oceanographic changes indicated in the bars: 1-Clement et al., 1999; 2-Sandweiss et al., 2001; 3-Rein et al., 2005; 4-Koutavas and Joanides, 2012; 5-Cobb et al. 2013; 6-Carré et al., 2014; 7-Emile-Geay et al., 2016; 8-Leonard et al., 2016b; 9-Thompson et al., 2017; 10-Toth et al. 2015a; 11-McGregor et al., 2013; 12-Toth et al., 2015b; 13-Conroy et al., 2008; 14-Corrège et al. 2000; 15-Haug et al. 2001; 16-Moy et al., 2002.**





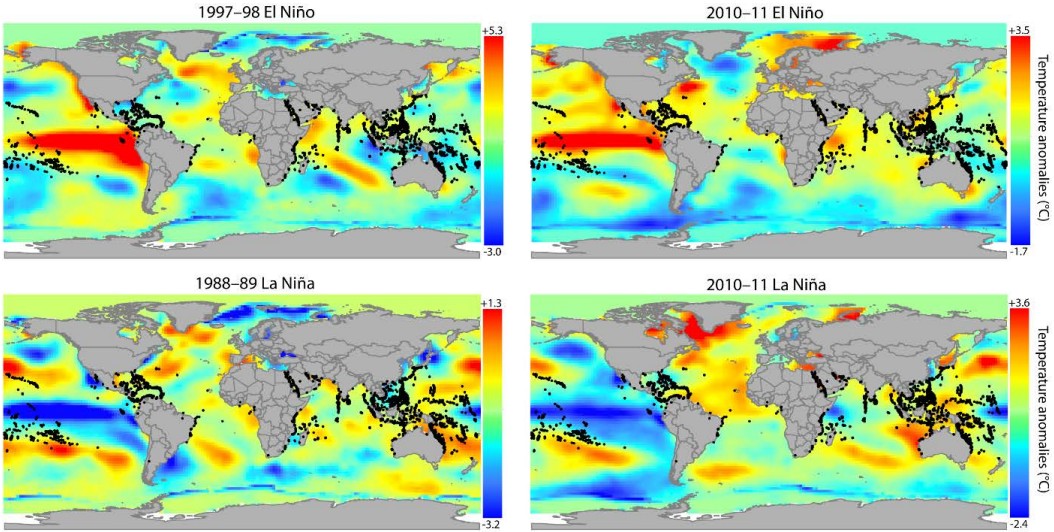

**Figure 4: Thermal anomalies associated with two of the strongest El Niño events (1997–98 and 2015–16) and La Niña events (1988–89 and 2010–11) in recent history. Black points on the maps indicate the locations of**
**coral reefs. The imagery was created with data from NOAA's Smith and Reynolds Extended, Reconstructed Sea-Surface Temperature, Level 4 Monthly Dataset (V5) using NASA's PO.DAAC LAS V8.6.1 visualization tool (https://podaac-tools.jpl.nasa.gov/las/).**