# Peer review of "The 4.2-ka event, ENSO, and coral-reef development"

_Climate of the Past, 2018_

## Referee Comment (RC1) · Anonymous Referee #1 · 6 Nov 2018

1. The authors stated a hypothesis that changes in ENSO activity around 4.2 ka triggered coral reef shutdown for about 2500 years (roughly between 4.1 and 1.6ka) in the Eastern Pacific. The hypothesis involved some hot topics including the late Holocene ENSO, 4.2 ka event and reef coral bleaching and mortality, therefore it sounds very interesting, but the authors did not provide direct evidences to support their hypothesis. Apart from the hypothesis itself, the authors did not provide any new information. 2. The basis of the authors' hypothesis is the mentioned "hiatus", i.e. the vertical accretion ceased from ∼4100 to 1600 cal BP (totally 2500 years) in their reef cores. On one hand, the authors did not show the detailed information about their cores, such as the reef type (fringing reef, atoll, barrier reef), the spatial distribution and the lengths of the cores. On the other hand, the authors did not tell us whether the reef also ceased

the development laterally. Most likely, their reef changed development orientation from vertical to lateral, because of sea level oscillations. If so, such change or the 2500-year hiatus should be controlled by sea level oscillation, rather than the 4.2 ka climate and the related ENSO activities. In this case, their hypothesis is wrong. 3. Modern observations have suggested that the large-scale coral bleaching and mortality were mostly associated with strong El Niño events, which exert high levels of thermal stress to the corals. This study, however, suggested the attenuated ENSO variability and the La Niña conditions in the 4.2-ka event had suppressed coral populations, and leaded to the shutdown of the reef accretion. The logic seems inconsistent with the modern observations. 4. Table 1 shows the time range of the beginnings of the hiatus are wide (from 5 to 3.8 ka BP), and it is not strictly around ∼4.2 ka BP, which suggests the reef hiatus was not related to the ∼4.2 ka event. 5. Based on the high $\Delta R$ and the low Sr/Ca-SST (Fig. 2), the authors suggested that strong upwelling occurred in ∼3.8-3.6 ka BP and partially attributed the hiatus to the upwelling. However, the variations of the $\Delta R$ and the Sr/Ca-SST are not always in phase, particularly for the last millennium. Could the authors clarify the relationship between the upwelling and the SST? 6. It is well known that ENSO variability has been closely linked with the strength of Easter Asian Summer monsoon throughout the Holocene. The Asian stalagmites, which recorded the evolution history of East Asian Summer monsoon with precise dating controls, documented the 4.2 ka events lasting only hundreds of years. However, the authors claimed that there existed a 2500-year shut down of vertical reef accretion in the tropical Eastern Pacific beginning 4.2 ka, and tied to increased variability of ENSO. 7. If the hypothesis is correct, the ENSO plays a role in climate change at 4.2 ka. What are the ultimate causes driving ENSO variability? 8. The Asian monsoon is generally suppressed during El Nino events and enhanced during La Nina events. According to the Figure 3, during the hiatus period, the tropical ocean experienced different ENSO modes, but the climate in Asian monsoon areas experienced a dry period during the 4.2 ka event. How to explain it? 9. Tectonic activity is also a possible cause to result in the stagnate of coral reef accretion vertically.

---

## Referee Comment (RC2) · Anonymous Referee #2 · 13 Nov 2018

This manuscript sets out to explore the relationship between an accretion hiatus in a Panamanian coral reef (and reef growth hiatuses in other locations) and the 4.2ka event. The manuscript puts forward that the two are linked via changes in mid to late Holocene El Niño-Southern Oscillation (ENSO) variability. The co-timing and possible inter-relatedness of the Panama reef growth hiatus and the 4.2ka event is an intriguing possibility, however the narrow focus of the manuscript on ENSO as the cause is problematic. As the manuscript discusses, two out of the three major El Niño events of the past 40 years did not result in major mass coral death in the tropical eastern Pacific, La Niña events may or may not also lead to coral death in the region, and the relationship may be indirect (e.g. via Acanthaster outbreaks). Furthermore there is mixed evidence that there has even been a change in ENSO over the time period

discussed. Together, this makes it is difficult to attribute the reef growth hiatus-4.2ka event co-timing to ENSO. The manuscript then extrapolates the reef growth hiatus-4.2ka event-ENSO links to explain other Pacific coral reefs. This is a stretch, especially without a more rounded discussion of the various factors that can influence reef growth at those locations. Overall, the manuscript needs to rebalance and expand the discussion to look at what non-ENSO factors (e.g. sea level, others) or ENSO-related factors (e.g. SST gradients, others) could also explain the reef growth hiatuses in the eastern Pacific and beyond.

Major points expanding on, and in addition to, those described above

1. Present day ENSO impacts on tropical eastern Pacific reefs. The discussion presented in section 2.2. 'Response to ENSO events' is well written and interesting, however it does highlight one of the key issues with the manuscript. That is, that the relationship between ENSO and coral death is complicated. Large amplitude El Niño events have not universally resulted in mass bleaching (e.g. p5 149-169) and sometime impacts are indirect. It implies that only large El Niño events have an impact on reef accretion, but what about moderate events? Conversely, La Niña apparently can also have a negative effect on coral reef growth in this region (e.g. p7 241-247), however the effects of La Niña are not discussed in this section. A broader discussion of ENSO impacts is needed.

2. Abrupt transition. p8, 255-256 the manuscript describes an abrupt transition to cooler and wetter conditions. However, looking at Figure 2 there is no data from ∼4.3 to 3.9ka so we do not know whether the transition was abrupt or not. The language around the commencement of the reef accretion hiatus needs to be toned down.

Related, why are there coral records through the hiatus? Presumably some corals survived. This point may have been explained by the authors in earlier papers but some explanation should be included in this manuscript.

Figure 2 implies that the Sr/Ca and d18O records are continuous over the intervals

where data is available, which they do not seem to be in earlier publications. The full coral data points should be presented.

Also in Figure 2, the x-axis break is misleading because some of the Sr/Ca and d18O data overlap with the axis break — the data should be plotted on the full x-axis.

3. Mid to late Holocene ENSO evidence. The study concludes that ENSO was the prime driver of the reef growth collapse, and presents a schematic of a sequence of ENSO-related changes (Fig. 3) to support their conclusions. The issue is that the sequence of events is not as clear as the manuscript present and some evidence has not been included. For example, p8, 261-264 cites Corrége et al. (2000) as evidence of ENSO but similar-aged results from Bayes Islet, presented in Emile-Geay et al. (2016), should also be included. Furthermore, the manuscript includes data from Moy et al. (2002), however there is some controversy over whether this record solely reflects El Niño events (e.g. Rodbell et al. QSR 2008; Emile-Geay and Tingley 2016). And, it could well be argued that reduced ENSO variability was established several centuries to a millennia before Panamá reef death (e.g. summarised in Emile-Geay et al. 2016). If ENSO variability was reduced in the centuries before the Panamá reef growth hiatus, is there really a link between ENSO and the reef growth?

Furthermore, p8, 274-282 presents an argument relating the waxing and waning of ENSO variability working to suppress and then initiate reef accretion, and the discussion presents the ENSO literature as if this pattern of variability is well established. However, as the manuscript states on p9, 290-302 discusses that the pattern of ENSO variability over the mid to late Holocene is far from clear. This further makes it very difficult to attribute the reef accretion hiatus and re-establishment to ENSO.

There are also some discrepancies in describing mid to late Holocene ENSO. For example, p10, 356-357 manuscript states "increased ENSO variability 4.2ka". However, earlier in the manuscript 4.2ka is described as having low ENSO variability (and as illustrated in the Figure 3 schematic). The descriptions of ENSO at 4.2ka, before and

after is inconsistent throughout, complicated by differences in the evidence in the literature.

Overall a more nuanced discussion of ENSO over the mid to late Holocene is required and the possibility that there is no link, that a link cannot be determined at this time, or that a link may be indirect needs to be discussed.

4. Other explanations. The mechanistic link between ENSO and reef death is not fully explained. The link in the specific examples for recent intense events points to bleaching or crown of thorns outbreak but this is not a consistent e.g. large events didn't necessarily cause reef death. Overall I would suggest that at this point it is very difficult to conclude that ENSO led to reef death event at 4.2ka.

While ENSO can't be ruled out (point 3) the manuscript should go further and discuss alternative explanations for the reef growth hiatus. For example, could changes be related to changes in the Pacific tropical SST gradient that could influence thermocline depth and upwelling strength (e.g. White et al. 2018 and papers therein). Or monsoon-driven changes in trade wind strength (which would also affect SST gradients and upwelling). Or what about sea level changes? Other possibilities?

5. Reef accretion changes elsewhere in the Pacific. p11, 360-370 and Table 1 The discussion of global reef perturbation events misses the important paper of Dechnik et al. (2018, recent but also earlier Dechnik papers) and related studies. The results of Dechnik et al. (2018) should be included since there is a potential for ENSO impacts via the teleconnection to the GBR. But really, most of the 'Prospectus' section is speculative and the manuscript would be better to consider a wider range of drivers for reef perturbation events (similar to the approach of Dechnik et al. 2018). Indeed, as I pointed out above, after devoting much of the manuscript to trying to attribute the Panamá reef growth hiatus to ENSO the second last paragraph of the 'Prospectus' only briefly discusses that there is strong evidence from the central Pacific that there has been very little ENSO variations over the mid to late Holocene. A broader discussion

of reef accretion hiatuses at other locations would be more balanced.

Minor points

p3, 75 The citations for the final sentence imply that these publications were the foundation papers for the use of corals to reconstruct past environmental and climate conditions. This is not the case and the citations should be revised.

p3, 80 ENSO impacts are not necessarily felt most keenly by marine ecosystems. Drought, for example, can have equally devastating impacts on terrestrial ecosystems.

p3, 100 Why and how did Wood et al. (2016) "cast considerable doubt on Richmond's proposed oceanographic teleconnection between the central and eastern Pacific".

p4, 115 Not clear how this paragraph ties into the discussion in section 2.1. Please clarify.

p4, 125 Confusing. Panama mortality levels are described as "intermediate" but given the percent mortality quoted I would have thought the mortality levels were high. Also, why is the upwelling/non-upwelling state of the Gulf of Panamá and Gulf of Chiriquí important if the bleaching was due to thermal anomalies associated with the 1982/1983 El Niño. Perhaps further discussion of upwelling impacts is warranted (along with major points above).

p5, 145 states that the 1997/1998 and 1982/1983 El Niño events were "enhanced by global warming" but has this really been established? At the very least citations need to be given to justify this statement.

p5, general comment - a map of the location of the various islands and Gulfs discuss in the text would be useful.

p5, 225-226 The "earlier model of Glynn (2000)" should be explained.

p7, 229-231 The modelling papers cited here are not evidence of mid-Holocene low ENSO variability. Remove. Tudhope et al. 2001 should be cited.

p7, 235 I would be wary of citing Leonard et al. (2016) here as an ENSO signal because records from the GBR reflect the teleconnection between ENSO and the climate of the GBR, not ENSO variability itself.

p7, 239-241 This sentence refers to the 1997/1998 El Niño leading to "widespread coral mortality", however this seems to contradict statements in the paragraph beginning on p5 149.

Figure 4 appears to have the wrong label for the upper right map (should be 2015-16?).

---

## Referee Comment (RC3) · Evan Edinger (Referee) · 21 Nov 2018

Summary: This is a discussion paper, rather than a paper in which new data are presented. This discussion paper summarizes the authors' research on the impacts of El-Nino Southern Oscillation variability on the coral reefs of Pacific Panama. The Pacific coast of Panama is a particularly suitable location to study this question, because it hosts coral reefs in settings that are affected by upwelling (Gulf of Panama) and not affected (Gulf of Chiriqui). The summary of those impacts allows the authors to propose that tropical, rather than high-latitude, climatic shifts may have been responsible for triggering the climate shift at 4.2 ka that marks the transition from the middle to the late Holocene. The authors summarize the oceanography, ecology, and paleoceanography of Tropical Eastern Pacific coral reefs en route to proposing a tropical driver for

this event.

Evaluation: The authors' arguments are based upon well-supported data in their region, and seem to make sense from the perspective of Pacific Panama. What the manuscript lacks is a broader geographic perspective on paleoecological changes in tropical coral reefs around the 4.2 ka event. The authors present maps of sea surface temperature anomalies during two El-Nino and two La-Nina events between 1988 and 2011, showing the distribution of thermal anomalies in relation to the distribution of coral reefs. The authors also present a table listing other regions with reported perturbations to coral reef systems (broadly) around 4.2 ka., but do not discuss these at great length. The next step, perhaps too ambitious for a discussion paper like this one, is to compare these changes, and also to compile examples of Pacific and Indian Ocean tropical coral reefs that were not affected by climatic changes at that time, to help decipher the global climatic changes affecting tropical coral reefs at 4.2 ka. While some of these data have already been compiled (see for example chronologies of Holocene reef growth in Montaggioni 2005, and Montaggioni & Braithwaite 2009), they still need to be compared to make a global analysis of synchronous climatic impacts on coral reefs at that time. For example, many of the changes to Holocene reef growth rates can be modulated by sea level and accommodation space, rather than by climatic change. Other explanations in particular cases relate to wave climate, apparently tied to ENSO (Roomey et al 2004, cited by the authors). Hence, the paper largely serves its purpose, to stimulate discussion and point to new research directions, but it does not conclusively demonstrate a tropical, specifically ENSO-related, driver to the 4.2ka shift in climate.

References: Montaggioni, L., 2005. History of Indo-Pacific coral reef systems since the last glaciation: development patterns and controlling factors. Earth Science Reviews 71: 1-75.

Montaggioni, LF., Braithwaite, CJR, Quaternary coral reef systems: development processes and controlling factors. Developments in Marine Geology 5.

---

## Author Comment (AC1) · 27 Nov 2018

Toth and Aronson Response to Reviewer #1

1. The authors stated a hypothesis that changes in ENSO activity around 4.2 ka triggered coral reef shutdown for about 2500 years (roughly between 4.1 and 1.6ka) in the Eastern Pacific. The hypothesis involved some hot topics including the late Holocene ENSO, 4.2 ka event and reef coral bleaching and mortality, therefore it sounds very interesting, but the authors did not provide direct evidences to support their hypothesis. Apart from the hypothesis itself, the authors did not provide any new information.

***Our aim in this review paper is to present the hypothesis that ENSO could have played a role in the 4.2 ka event and outline a framework for how this hypothesis could be tested using records from coral reefs in the tropical Pacific. It was not our goal to provide new data or information, but rather to review the existing literature in the context of our hypothesis. We reworded the end of our introduction to clarify this point:***
*Whereas the majority of the records of the 4.2-ka event have come from terrestrial environments, the contemporary impacts of ENSO are often felt most keenly in marine ecosystems. Understanding whether and how marine ecosystems responded to climatic changes around 4.2 ka is, therefore, critical to deciphering the ultimate drivers of the 4.2-ka event. Here, we explore the hypothesis that ENSO played a role in the 4.2 ka event by reviewing paleoecological and paleoceanographic records from marine environments in the tropical Pacific. We focus on the long-term collapse of coral-reef development in the tropical eastern Pacific (TEP) to evaluate the role of which does appear to be related to changes in ENSO in the 4.2-ka event. The conclude that the relationship between ENSO and the 4.2 ka event warrants further study and outline a conception framework for future studies to investigate the linkages between these climatic phenomena using records from coral-reef environments in the tropical Pacific.*

2. The basis of the authors' hypothesis is the mentioned "hiatus", i.e. the vertical accretion ceased from ~4100 to 1600 cal BP (totally 2500 years) in their reef cores. On one hand, the authors did not show the detailed information about their cores, such as the reef type (fringing reef, atoll, barrier reef), the spatial distribution and the lengths of the cores. On the other hand, the authors did not tell us whether the reef also ceased the development laterally. Most likely, their reef changed development orientation from vertical to lateral, because of sea level oscillations. If so, such change or the 2500-year hiatus should be controlled by sea level oscillation, rather than the 4.2 ka climate and the related ENSO activities. In this case, their hypothesis is wrong.

***We have clarified that all the reefs in Pacific Panamá are fringing reefs (there are no barrier reefs in the eastern Pacific and atolls are uncommon). Detailed information about the cores is provided in previous publications and we now include a reference to where that information can be found. The reasons why sea-level variability is unlikely to have caused the hiatus have been discussed in previous publication. We have added the following sentence to point out those studies:***
*The fact that the the shutdown in reef accretion occurred in both the Gulf of Panamá and the Gulf of Chiriquí excludes upwelling and outbreaks of Acanthaster as drivers: upwelling is weak or absent in the Gulf of Chiriquí, and Acanthaster is absent from the Gulf of Panamá. Other possible factors, including changes in relative sea level, tectonics, and bioerosion, cannot explain the observed patterns either (Toth et al., 2012, 2015a).*

***We did add some additional information about the depths where the cores were collected and where the hiatus occurred; however, the sea-level history of the eastern Pacific is unclear so we are unable to convert those data to paleodepths. Nonetheless, the broad range of depths over which the hiatus occurs (~1.6 m) suggests that lateral accretion was also limited at the onset of the hiatus in reef accretion. The description of the core records now reads:***

*Our push-cores from three fringing Panamanian reefs, which spanned across a gradient of upwelling and water depths (-0.8 to -4.1 m relative to mean sea level), and which we dated with radiocarbon and uranium-series techniques, showed that vertical accretion essentially ceased from ~4100 to 1600 cal BP (Fig. 2; calibrated calendar years before 1950; Toth et al., 2012). There is no evidence that the reefs shifted to lateral reef accretion at this time, as the reefs were growing in a broad range of depth environments (~1.6 m across the cores included in Toth et al. [2012]) at the time of shutdown. The hiatus in growth lasted approximately 2500 years, meaning the reefs were in a phase of negligible growth for as much as 40% of their history (Toth et al., 2012). Detailed core logs are provided in Toth et al. (2012, 2013).*

3. Modern observations have suggested that the large-scale coral bleaching and mortality were mostly associated with strong El Niño events, which exert high levels of thermal stress to the corals. This study, however, suggested the attenuated ENSO variability and the La Niña conditions in the 4.2-ka event had suppressed coral populations, and leaded to the shutdown of the reef accretion. The logic seems inconsistent with the modern observations.

***Our reconstructions suggest that enhanced La Niña likely provided the initial trigger for reef shutdown in Pacific Panamá; however, we hypothesized higher ENSO variability** overall (i.e., both La Niña and El Niño) suppressed reef development in the eastern Pacific for the next 2500 years. We added some text to clarify this point in the prospectus.*

4. Table 1 shows the time range of the beginnings of the hiatus are wide (from 5 to 3.8 ka BP), m and it is not strictly around ~4.2 ka BP, which suggests the reef hiatus was not related to the ~4.2 ka event.

***This is an interesting comment. The age-range for the start of the hiatus arises for several reasons. First is the uncertainty around dates in the cores. Second is the very-real possibility that coral branch-fragments were missing of the exact ages to delimit the actual start of the hiatus. Regarding the latter point, if 4200-year-old branch-fragments in excellent taphonomic condition, which denote rapid population growth, are missing from the particular spot where we cored in Panamá, or where someone else sampled in another locality, then the estimated start-time of the hiatus will be artificially early. If, on the other hand, the oldest fragments from during the actual hiatus are missing from that particular spot, our estimate of the start-time will be artificially late. A third point worth noting is that we do not yet fully comprehend how the 4.2-ka event was manifested on Pacific reefs and how it interacted with changes in relative sea level. These points are now articulated in the first paragraph of Section 5 of the paper:***

*however, falls or stillstands in relative sea-level in the western Pacific after the middle Holocene likely also contributed to stalled late-Holocene reef growth in many locations (e.g., Dechnik et al., 2018), potentially making it more difficult to discern the impacts of climatic variability and also potentially introducing variability into the start-times of hiatuses in reef accretion. Other possible causes of variations in the start-times listed in Table 1 include the artifactual absence of*

*coral material of particular ages in particular samples and the uncertainty associated with the dating techniques.*

5. Based on the high ΔR and the low Sr/Ca-SST (Fig. 2), the authors suggested that strong upwelling occurred in ~3.8-3.6 ka BP and partially attributed the hiatus to the upwelling. However, the variations of the ΔR and the Sr/Ca-SST are not always in phase, particularly for the last millennium. Could the authors clarify the relationship between the upwelling and the SST?

***We do not attribute the hiatus to upwelling, as the hiatus also occurred in the non-upwelling Gulf of Chiriquí. Instead, we use our record of upwelling to infer broader-scale changes in climate at this time, as upwelling is intensified during La Niña and suppressed during El Niño. The late Holocene changes in upwelling and SST provide further support for our argument about the changes in ENSO that may have allowed reef development to resume ~1600 years ago. We have added the following sentences to emphasize this point:***

*Our records suggest that upwelling was more moderate after the hiatus (Fig. 2A), which would indicate that there were fewer or less extreme La Niña events at this time. Furthermore, because the climate remained relatively cool after the hiatus (Fig. 2B), it is likely that the influence of El Niño events had also decreased.*

6. It is well known that ENSO variability has been closely linked with the strength of Easter Asian Summer monsoon throughout the Holocene. The Asian stalagmites, which recorded the evolution history of East Asian Summer monsoon with precise dating controls, documented the 4.2 ka events lasting only hundreds of years. However, the authors claimed that there existed a 2500-year shut down of vertical reef accretion in the tropical Eastern Pacific beginning 4.2 ka, and tied to increased variability of ENSO.

***This issue is addressed in the following sentence in Section 4:***

*Whereas the 4.2-ka event was manifested as an abrupt, short-lived climatic event in many locations, the large-scale climatic and ecosystem responses to the event may have been more gradual and more protracted.*

7. If the hypothesis is correct, the ENSO plays a role in climate change at 4.2 ka. What are the ultimate causes driving ENSO variability?

***The ultimate causes of ENSO variability are still being debated, as we discuss in the final paragraph of Section 4.***

8. The Asian monsoon is generally suppressed during El Nino events and enhanced during La Nina events. According to the Figure 3, during the hiatus period, the tropical ocean experienced different ENSO modes, but the climate in Asian monsoon areas experienced a dry period during the 4.2 ka event. How to explain it?

***We agree that most of the records of the 4.2 ka event reflect El Niño-like, as we discuss in the second paragraph of Section 4; however, we also point out in the following paragraph that some records indicate that more La Niña-like conditions followed this period. We added a reference that makes this suggestion in relation to the east Asian monsoon:***

*Similarly, whereas a number of records indicate a weaker East Asian Monsoon around 4.2 ka, indicative of an El Niño-like climate (Straubwasser et al., 2003), there is also evidence that some regions of southern Asia became wetter after 4.2 ka (reviewed in Wu and Liu, 2004).*

9. Tectonic activity is also a possible cause to result in the stagnate of coral reef accretion vertically.

*Please see our response to Comment 2.*

---

## Author Comment (AC2) · 27 Nov 2018

Toth and Aronson Response to Reviewer #2

This manuscript sets out to explore the relationship between an accretion hiatus in a Panamanian coral reef (and reef growth hiatuses in other locations) and the 4.2ka event. The manuscript puts forward that the two are linked via changes in mid to late Holocene El Niño-Southern Oscillation (ENSO) variability. The co-timing and possible inter-relatedness of the Panama reef growth hiatus and the 4.2ka event is an intriguing possibility, however the narrow focus of the manuscript on ENSO as the cause is problematic.
***Please see our responses below for treatment of other causes besides ENSO.***

As the manuscript discusses, two out of the three major El Niño events of the past 40 years did not result in major mass coral death in the tropical eastern Pacific, La Niña events may or may not also lead to coral death in the region, and the relationship may be indirect (e.g. via Acanthaster outbreaks).
***We exclude* Acanthaster *outbreaks as a cause in section 3. Also, please see our responses below under the reviewer's point 1 for discussion of both La Niña and the variability of responses to ENSO events.***

Furthermore there is mixed evidence that there has even been a change in ENSO over the time period discussed. Together, this makes it is difficult to attribute the reef growth hiatus-4.2ka event co-timing to ENSO.
***The majority of the ENSO literature supports the conclusion that there was an increase in ENSO variability after 4.2 ka and our records from Pacific Panamá suggest that, at least in the eastern Pacific, those changes coincided with the shutdown of reef accretion. This evidence provides the basis of our hypothesis that there may be a connection between the 4.2-ka event and ENSO and we hope that this hypothesis will be tested by other researchers in the future. We Please also see our response to comment 3.***

The manuscript then extrapolates the reef growth hiatus-4.2ka event-ENSO links to explain other Pacific coral reefs. This is a stretch, especially without a more rounded discussion of the various factors that can influence reef growth at those locations. Overall, the manuscript needs to rebalance and expand the discussion to look at what non-ENSO factors (e.g. sea level, others) or ENSO-related factors (e.g. SST gradients, others) could also explain the reef growth hiatuses in the eastern Pacific and beyond.
***We have framed our ideas more clearly as hypotheses. As remarked above and in our response to Comment 4, we have also treated other possible causes more thoroughly than in our previous version of the manuscript.***

1. Present day ENSO impacts on tropical eastern Pacific reefs. The discussion presented in section 2.2. 'Response to ENSO events' is well written and interesting, however it does highlight one of the key issues with the manuscript. That is, that the relationship between ENSO and coral death is complicated. Large amplitude El Niño events have not universally resulted in mass bleaching (e.g. p5 149-169) and sometime impacts are indirect. It implies that only large El Niño events have an impact on reef accretion, but what about moderate events? Conversely, La Niña apparently can also have a negative effect on coral reef growth in this region (e.g. p7 241-247), however the effects of La Niña are not discussed in this section. A broader discussion of ENSO impacts is needed.

*We have reorganized the text related to modern ENSO impacts to emphasize the role of La Niña. The section describing the impacts of La Niña in Pacific Panamá has been moved from Section 3 to Section 2.2 where the impacts of El Niño in Pacific Panamá are discussed:*
*La Niña is also problematic for corals in Pacific Panamá, as lowered sea level in the TEP during La Niña events causes more frequent coral mortality associated with subaerial exposure (Eakin and Glynn, 1996; Toth et al., 2017). In addition, La Niña is associated with elevated rainfall in Pacific Panamá, which increases turbidity, and enhanced upwelling, which reduces water temperatures, decreases pH, and increases nutrient levels, all of which act to suppress coral growth (Glynn, 1976).*

*The variability of ENSO impacts is an area of active research, and a detailed review of that topic is beyond the scope of this paper.*

2. Abrupt transition. p8, 255-256 the manuscript describes an abrupt transition to cooler and wetter conditions. However, looking at Figure 2 there is no data from ~4.3 to 3.9ka so we do not know whether the transition was abrupt or not. The language around the commencement of the reef accretion hiatus needs to be toned down.
*We do have data from our record of ΔR during this period, which suggests a rapid increase in upwelling, indicative of intensifying La Niña-like conditions; however, we removed the word "abrupt".*

Related, why are there coral records through the hiatus? Presumably some corals survived. This point may have been explained by the authors in earlier papers but some explanation should be included in this manuscript.
*We have added a description of these samples to the paragraph where we begin to describe our records. The beginning of that paragraph now reads:*

*We tested the hypothesis that changes in ENSO activity around ~4.2 ka triggered reef shutdown in Pacific Panamá by evaluating geochemical proxy records from corals in our cores (Fig. 2; Toth et al., 2015a, 2015b). For obvious reasons, the availability of coral samples from during the hiatus was limited; however, we were able to analyze the Sr/Ca and $\delta^{18}O$ of seven coral skeletons, which our age model suggested grew at the beginning of the hiatus (~3.9–3.6 ka; Toth et al. 2015a). We also have a measurement of oceanic radiocarbon, a proxy for ocean circulation, from one coral that grew during this period.*

Figure 2 implies that the Sr/Ca and d18O records are continuous over the intervals where data is available, which they do not seem to be in earlier publications. The full coral data points should be presented.
*We have added information to the caption for Figure 2, which clarifies that the Sr/Ca and d18O data are presented as 200-year running means of the raw data:*

*(B & C) provide reconstructions of temperature and salinity variability for Pacific Panamá based on 200-yr running means (±95% CIs) of Sr/Ca and δ18O of corals sampled from a core collected from the reef at Contadora Island in the Gulf of Panama. The individual data points used to generate the curves can be found in Toth et al. (2015a) and Toth (2013).*

Also in Figure 2, the x-axis break is misleading because some of the Sr/Ca and d18O data overlap with the axis break ã˘AˇT the data should be plotted on the full x-axis

***Removing the axis break would require condensing the data in the figure to the point that it would be difficult for readers to discern the pertinent trends we discuss. We do not present any Sr/Ca or d18O data during the hiatus after the axis break (at 3.4 ka), so we are not sure what the reviewer is referring to. We did switch the direction of the axis break lines, which we hope will make this clearer.***

3. Mid to late Holocene ENSO evidence. The study concludes that ENSO was the prime driver of the reef growth collapse, and presents a schematic of a sequence of ENSO-related changes (Fig. 3) to support their conclusions. The issue is that the sequence of events is not as clear as the manuscript present and some evidence has not been included. For example, p8, 261-264 cites Corrége et al. (2000) as evidence of ENSO but similar-aged results from Bayes Islet, presented in Emile-Geay et al. (2016), should also be included. Furthermore, the manuscript includes data from Moy et al. (2002), however there is some controversy over whether this record solely reflects El Niño events (e.g. Rodbell et al. QSR 2008; Emile-Geay and Tingley 2016). And, it could well be argued that reduced ENSO variability was established several centuries to a millennia before Panamá reef death (e.g. summarised in Emile-Geay et al. 2016). If ENSO variability was reduced in the centuries before the Panamá reef growth hiatus, is there really a link between ENSO and the reef growth? Furthermore, p8, 274-282 presents an argument relating the waxing and waning of ENSO variability working to suppress and then initiate reef accretion, and the discussion presents the ENSO literature as if this pattern of variability is well established. However, as the manuscript states on p9, 290-302 discusses that the pattern of ENSO variability over the mid to late Holocene is far from clear. This further makes it very difficult to attribute the reef accretion hiatus and re-establishment to ENSO. There are also some discrepancies in describing mid to late Holocene ENSO. For example, p10, 356-357 manuscript states "increased ENSO variability 4.2ka". However, earlier in the manuscript 4.2ka is described as having low ENSO variability (and as illustrated in the Figure 3 schematic). The descriptions of ENSO at 4.2ka, before and after is inconsistent throughout, complicated by differences in the evidence in the literature. Overall a more nuanced discussion of ENSO over the mid to late Holocene is required and the possibility that there is no link, that a link cannot be determined at this time, or that a link may be indirect needs to be discussed.

***We agree that evolution of ENSO over the mid- to late Holocene is a complex issue that has not been fully resolved in the literature. We discuss some of these limitations at the end of section 3, but we have modified the text throughout this section to make the uncertainty in past ENSO more apparent. The reviewer is correct that the majority of the paleoclimate literature suggests that ENSO variability was low in the mid-Holocene and we discuss this clearly in the beginning of section 3. The literature also suggests, however, that ENSO variability increased after this time, which we hypothesized provided the trigger for reef shutdown in the eastern Pacific. To acknowledge that our schematic (Fig. 3) cannot fully represent these complexities, we added the following qualifying statement before our discussion of how ENSO may relate to the shutdown of reef development in Pacific Panama:***

*The schematic provides a simplified summary of the changes in ENSO suggested by our study and the paleoclimate literature, which could have contributed to the shutdown of eastern Pacific reef development and is not meant to provide a comprehensive review of the literature.*

*As suggested, we have added a reference to the new record from Bayes Inlet:*
*At least one record from the western Pacific (Vanuatu) indicates that a multi-decadal period of enhanced ENSO variability followed just 500 years after the putative low in ENSO activity at ~4.2 ka (Corrège et al., 2000); however, there is less evidence for elevated ENSO variability in a recent contemporaneous record from nearby New Caledonia (Emile-Geay et al., 2016).*

*We have also added a statement qualifying the Moy record based on new analyses:*
*...a series of records from the same region suggest a possible peak in the frequency of El Niño events around 3 ka (Sandweiss et al., 2001; Moy et al., 2002); however, recent reanalysis has called this conclusion into question (Emile-Geay and Tingley 2016).*

*Finally, we have corrected the statement at the beginning of the prospectus (formerly p10, 356-357) to: "...increased ENSO variability* under *after* *4.2 ka"*

4. Other explanations. The mechanistic link between ENSO and reef death is not fully explained. The link in the specific examples for recent intense events points to bleaching or crown of thorns outbreak but this is not a consistent e.g. large events didn't necessarily cause reef death. Overall I would suggest that at this point it is very difficult to conclude that ENSO led to reef death event at 4.2ka. While ENSO can't be ruled out (point 3) the manuscript should go further and discuss alternative explanations for the reef growth hiatus. For example, could changes be related to changes in the Pacific tropical SST gradient that could influence thermocline depth and upwelling strength (e.g. White et al. 2018 and papers therein). Or monsoon driven changes in trade wind strength (which would also affect SST gradients and upwelling). Or what about sea level changes? Other possibilities?

*We have added language to Section 3 discussing and excluding other causal explanations, including sea level, tectonics, and bioerosion:*
*The fact that the the shutdown in reef accretion occurred in both the Gulf of Panamá and the Gulf of Chiriquí excludes upwelling and outbreaks of Acanthaster as drivers: upwelling is weak or absent in the Gulf of Chiriquí, and Acanthaster is absent from the Gulf of Panamá. Other possible factors, including changes in relative sea level, tectonics, and bioerosion, cannot explain the observed patterns either (Toth et al., 2012, 2015a).*

5. Reef accretion changes elsewhere in the Pacific. p11, 360-370 and Table 1 The discussion of global reef perturbation events misses the important paper of Dechnik et al. (2018, recent but also earlier Dechnik papers) and related studies. The results of Dechnik et al. (2018) should be included since there is a potential for ENSO impacts via the teleconnection to the GBR. But really, most of the 'Prospectus' section is speculative and the manuscript would be better to consider a wider range of drivers for reef perturbation events (similar to the approach of Dechnik et al. 2018). Indeed, as I pointed out above, after devoting much of the manuscript to trying to attribute the Panamá reef growth hiatus to ENSO the second last paragraph of the 'Prospectus' only briefly discusses that there is strong evidence from the central Pacific that there has been very little ENSO variations over the mid to late Holocene. A broader discussion of reef accretion hiatuses at other locations would be more balanced.

*This is a good point. We have adopted this suggestion, while remaining within the scope of the manuscript, by adding to the Prospectus language and references, including papers by Dechnik and Webster about the GBR.*

6. p3, 75 The citations for the final sentence imply that these publications were the foundation papers for the use of corals to reconstruct past environmental and climate conditions. This is not the case and the citations should be revised.
***We did not mean to imply that these were foundational or review papers, but rather examples of environmental reconstructions from coral reefs. We have added*** *"e.g.,"* ***before the references to more clearly reflect this.***

7. p3, 80 ENSO impacts are not necessarily felt most keenly by marine ecosystems. Drought, for example, can have equally devastating impacts on terrestrial ecosystems.
***We have added*** *"often"* ***before*** *"felt most keenly"* ***to qualify this statement.***

8. p3, 100 Why and how did Wood et al. (2016) "cast considerable doubt on Richmond's proposed oceanographic teleconnection between the central and eastern Pacific".
***The doubt was based on oceanographic modeling. We have clarified this point and added a second reference in support of the statement. The sentence now reads:***
*Oceanographic-modeling exercises, however, Wood et al. (2016), however, cast considerable doubt on Richmond's proposed oceanographic teleconnection between the central and eastern Pacific, at least so far as* Pocillopora *is concerned (Wood et al., 2016; Romero-Torres et al., 2018).*

9. p4, 115 Not clear how this paragraph ties into the discussion in section 2.1. Please clarify.
***This comment is spot-on. We have removed the paragraph from section 2.1 and placed the text where it fits better, in sections 2.2 and 2.3.***

10. p4, 125 Confusing. Panama mortality levels are described as "intermediate" but given the percent mortality quoted I would have thought the mortality levels were high.
***We changed this to*** *"somewhat lower".*

Also, why is the upwelling/non-upwelling state of the Gulf of Panamá and Gulf of Chiriquí important if the bleaching was due to thermal anomalies associated with the 1982/1983 El Niño. Perhaps further discussion of upwelling impacts is warranted (along with major points above).
***We introduce the upwelling regimes of the Gulfs here so that we can discuss how upwelling has interacted with El Niño in the past. Whereas upwelling was suppressed by the El Niño event in 1982–83, upwelling during the 1997–98 event buffered reefs in the Gulf of Panamá from thermal anomalies. We have added the following sentence to the end of the section about the impacts of the 1982–83 event to clarify this point:***
*The similarity in the level of mortality throughout Pacific Panamá reflects the fact that El Niño suppressed seasonal upwelling in the Gulf of Panama in 1982–83 so the level of warming was similar in both Gulfs (Glynn et al., 2001).*

11. p5, 145 states that the 1997/1998 and 1982/1983 El Niño events were "enhanced by global warming" but has this really been established? At the very least citations need to be given to justify this statement.
***The sentence now reads:*** *"Both events may have been enhanced by global warming…"* ***and we have added a reference (Hughes et al. 2018) to support this idea.***

12. p5, general comment - a map of the location of the various islands and Gulfs discuss in the text would be useful.

***Location maps are provided in several previous publications (Toth et al. 2012, 2015a, 2015b, 2017), and we would prefer not to duplicate those maps here. We did include a reference to where a regional map showing the locations of eastern Pacific reefs can be found at the beginning of the section in question:***
*(see Fig. 6.1 in Toth et al., 2017 for a map of locations where coral reefs occur)*

13. p5, 225-226 The "earlier model of Glynn (2000)" should be explained.
***This sentence now reads:***
*The tempo and causality of reef development in Pacific Panamá represent a vastly scaled-up version of the earlier model of Glynn and Colgan (1992; 2000), which suggested that poor reef development was a consequence of decadal-scale disturbance by El Niño.*

14. p7, 229-231 The modelling papers cited here are not evidence of mid-Holocene low ENSO variability. Remove. Tudhope et al. 2001 should be cited.
***We have removed the Clement et al. 1999 reference, which relates ENSO variability to millennial-scale changes in insolation, but does not directly speak to changes around the mid-Holocene. The Liu et al. 2014 modeling reference does, however support the idea of relatively low ENSO variability during the mid-Holocene (with variability increasing to present), so we have retained this reference. We also added a citation another modeling study (Zheng et al. 2008), which directly evaluated ENSO variability at 6 ka and we added the Tudhope reference, as suggested. Finally, we qualified the sentence to say that the mid-Holocene was*** *"likely"* ***a period of low ENSO variability.***

15. p7, 235 I would be wary of citing Leonard et al. (2016) here as an ENSO signal because records from the GBR reflect the teleconnection between ENSO and the climate of the GBR, not ENSO variability itself.
***Leonard et al. 2016 interpret their record as being reflective of ENSO variability; however, it is true that all paleoclimate reconstructions of ENSO are overprinted by local climate variability. We have, therefore, changed the language here to say*** *"some studies have suggested that there was exceptionally low ENSO variability between 4.3 and 4.2 ka."*

16. p7, 239-241 This sentence refers to the 1997/1998 El Niño leading to "widespread coral mortality", however this seems to contradict statements in the paragraph beginning on p5 149.
***The sentence on p7 now reads:***
*In Pacific Panamá, the elevated water temperatures and high irradiance (low cloud cover) associated with the strong El Niño events in 1982–83 and 1997–98 caused widespread coral bleaching, which was associated with mass coral mortality in the earlier event (Glynn et al., 2001).*

17. Figure 4 appears to have the wrong label for the upper right map (should be 2015-16?).
***We have corrected this typo.***

---

## Author Comment (AC3) · 27 Nov 2018

Toth and Aronson Response to Reviewer #3

Summary: This is a discussion paper, rather than a paper in which new data are presented. This discussion paper summarizes the authors' research on the impacts of El-Nino Southern Oscillation variability on the coral reefs of Pacific Panama. The Pacific coast of Panama is a particularly suitable location to study this question, because it hosts coral reefs in settings that are affected by upwelling (Gulf of Panama) and not affected (Gulf of Chiriqui). The summary of those impacts allows the authors to propose that tropical, rather than high-latitude, climatic shifts may have been responsible for triggering the climate shift at 4.2 ka that marks the transition from the middle to the late Holocene. The authors summarize the oceanography, ecology, and paleoceanography of Tropical Eastern Pacific coral reefs en route to proposing a tropical driver for this event.

Evaluation: The authors' arguments are based upon well-supported data in their region, and seem to make sense from the perspective of Pacific Panama. What the manuscript lacks is a broader geographic perspective on paleoecological changes in tropical coral reefs around the 4.2 ka event. The authors present maps of sea surface temperature anomalies during two El-Nino and two La-Nina events between 1988 and 2011, showing the distribution of thermal anomalies in relation to the distribution of coral reefs. The authors also present a table listing other regions with reported perturbations to coral reef systems (broadly) around 4.2 ka., but do not discuss these at great length. The next step, perhaps too ambitious for a discussion paper like this one, is to compare these changes, and also to compile examples of Pacific and Indian Ocean tropical coral reefs that were not affected by climatic changes at that time, to help decipher the global climatic changes affecting tropical coral reefs at 4.2 ka. While some of these data have already been compiled (see for example chronologies of Holocene reef growth in Montaggioni 2005, and Montaggioni & Braithwaite 2009), they still need to be compared to make a global analysis of synchronous climatic impacts on coral reefs at that time. For example, many of the changes to Holocene reef growth rates can be modulated by sea level and accommodation space, rather than by climatic change. Other explanations in particular cases relate to wave climate, apparently tied to ENSO (Roomey et al 2004, cited by the authors). Hence, the paper largely serves its purpose, to stimulate discussion and point to new research directions, but it does not conclusively demonstrate a tropical, specifically ENSO-related, driver to the 4.2ka shift in climate.

***We thank Evan for his positive view of the manuscript. His comment about sea level and other drivers of reef accretion echoes suggestions in the other reviews, and we agree. Accordingly, we have added language to Section 5 pointing out that the hiatus, putatively driven by ENSO variability, could have been modulated by other factors such as variations in relative sea level.***

References:
Montaggioni, L., 2005. History of Indo-Pacific coral reef systems since the last glaciation: development patterns and controlling factors. Earth Science Reviews 71: 1-75.
Montaggioni, LF., Braithwaite, CJR, Quaternary coral reef systems: development processes and controlling factors. Developments in Marine Geology 5.